# Iron Dyshomeostasis in COVID-19: Biomarkers Reveal a Functional Link to 5-Lipoxygenase Activation

**DOI:** 10.3390/ijms24010015

**Published:** 2022-12-20

**Authors:** Beatrice Dufrusine, Silvia Valentinuzzi, Sandra Bibbò, Verena Damiani, Paola Lanuti, Damiana Pieragostino, Piero Del Boccio, Ersilia D’Alessandro, Alberto Rabottini, Alessandro Berghella, Nerino Allocati, Katia Falasca, Claudio Ucciferri, Francesco Mucedola, Marco Di Perna, Laura Martino, Jacopo Vecchiet, Vincenzo De Laurenzi, Enrico Dainese

**Affiliations:** 1Department of Bioscience and Technology for Food Agriculture and Environment, University of Teramo, 64100 Teramo, Italy; 2Department of Innovative Technologies in Medicine and Dentistry, University “G. d’Annunzio” of Chieti-Pescara, 66100 Chieti, Italy; 3Center for Advanced Studies and Technology (CAST), University “G. d’Annunzio” of Chieti-Pescara, 66100 Chieti, Italy; 4Department of Pharmacy, University “G. d’Annunzio” of Chieti-Pescara, 66100 Chieti, Italy; 5Department of Medicine and Aging Science, “G. d’Annunzio” University of Chieti-Pescara, 66100 Chieti, Italy; 6Clinic of Infectious Diseases, S.S. Annunziata Hospital, 66100 Chieti, Italy; 7Pneumology Department, “SS Annunziata” Hospital, 66100 Chieti, Italy

**Keywords:** COVID-19, long-COVID, iron metabolism, 5-lipoxygenase, leukotriene B4, lipocalin 2

## Abstract

Coronavirus disease 2019 (COVID-19) is characterized by a broad spectrum of clinical symptoms. After acute infection, some subjects develop a post-COVID-19 syndrome known as long-COVID. This study aims to recognize the molecular and functional mechanisms that occur in COVID-19 and long-COVID patients and identify useful biomarkers for the management of patients with COVID-19 and long-COVID. Here, we profiled the response to COVID-19 by performing a proteomic analysis of lymphocytes isolated from patients. We identified significant changes in proteins involved in iron metabolism using different biochemical analyses, considering ceruloplasmin (Cp), transferrin (Tf), hemopexin (HPX), lipocalin 2 (LCN2), and superoxide dismutase 1 (SOD1). Moreover, our results show an activation of 5-lipoxygenase (5-LOX) in COVID-19 and in long-COVID possibly through an iron-dependent post-translational mechanism. Furthermore, this work defines leukotriene B4 (LTB4) and lipocalin 2 (LCN2) as possible markers of COVID-19 and long-COVID and suggests novel opportunities for prevention and treatment.

## 1. Introduction

Severe acute respiratory syndrome coronavirus 2 (SARS-CoV-2) infection, commonly known as COVID-19, has affected 221 countries and caused 6.327.547 deaths worldwide, according to data reported by the World Health Organization [1]. COVID-19 transmission occurs by virus contact with oral, nasal, or ocular mucosa, or via the inhalation of respiratory droplets released from an infected person [2,3,4]. The clinical progression of COVID-19 ranges from an asymptomatic condition to a severe multisystemic inflammatory syndrome (MIS) leading to death [5,6]. Mild symptoms may include fever, cough, sore throat, malaise, myalgias, anosmia, and ageusia [5,7,8]. Less common mild symptoms include anorexia, diarrhea, headache, and hemoptysis [5,9,10]. Infected patients with MIS develop severe symptoms such as dyspnea, pneumonia, and severe acute respiratory distress syndrome (ARDS) that may lead to other secondary infections and lethal complications [11,12]. Risk factors for COVID-19 severity include older age, chronic obstructive pulmonary disease, cardiovascular disease, type 2 diabetes mellitus, obesity, chronic kidney disease, immunocompromised state, and cancer [6,13]. In some cases, once recovered from COVID-19, patients develop persistent or new symptoms lasting weeks or months; this new syndrome is described as “long-COVID syndrome” [14,15]. Estimates of the prevalence of long-COVID are extremely variable. In addition, the available studies have reported a variable percentage (ranging from 10 to 80%) of patients who experience long-COVID with respect to specific symptoms [16,17,18,19,20,21]. The most common definition used for long-COVID is the presence of symptoms for more than 3 months after acute infection [16,17]. Subjects with long-COVID show a variable set of physical and mental symptoms, such as chronic fatigue, breathlessness, cardiovascular abnormalities, neurocognitive impairments, anxiety, and depression, persisting for months after the resolution of infection [15,18,19,22]. To date, the risk factors for experiencing the long-term effects of COVID-19 are not yet well established, and more studies are required. However, immunological differences such as lymphopenia [23,24,25] and elevated serum antibodies, as well as SARS-CoV-2-specific antibodies, are prevalent among patients with severe forms of COVID-19 and/or long-COVID [26,27,28]. Furthermore, recent studies have highlighted a set of immune modulators as possible specific diagnostic biomarkers, suggesting decreased serum cortisol levels as one of the most relevant predictors for severe COVID-19 and, in particular, for long-COVID [29,30,31].

In the host–pathogen functional interaction, iron has been shown to be crucial in controlling fundamental biological processes, including DNA/RNA synthesis, transcription, and adenosine triphosphate generation. Moreover, iron overload conditions have been reported in different viral infections [32,33]. Iron homeostasis is tightly controlled by a fine and complex mechanism comprising metal uptake, transport, release, and cellular storage, as well as efficient and finely modulated metal recycling, primarily from the hemoglobin (Hb) of senescent erythrocytes. Briefly, iron is absorbed in the enterocytes, transported in the blood by transferrin (Tf), and stored at a high concentration within the cells as a result of binding to ferritin protein [34,35,36]. In enterocytes, iron-recycling macrophages, and iron-storing hepatocytes, intracellular iron can be exported by the transporter ferroportin (FPN) [37,38]. Cellular iron efflux is modulated by hepcidin, an hepatic hormone that binds FPN, thus favoring an increase in intracellular iron and lowering plasma levels of iron [36,39,40,41]. Tf-iron transport is also modulated by a copper enzyme (i.e., ceruloplasmin, Cp) that induces iron binding to Tf with ferroxidase activity [37,42]. Thus, most cells can import iron by capturing iron-loaded Tf through its binding to Tf-receptors, with consequent internalization [43,44]. Other proteins considered as modulators of iron are lipocalin 2 (LCN2) (for its role in binding iron-loaded siderophores and controlling iron cellular content [45,46,47]) and the protein hemopexin (HPX), that binding free heme mainly lost via Hb recycling is (together with superoxide dismutase 1 (SOD1)) involved in cellular protection against heme-driven and iron-induced oxidative stress [48,49,50,51]. In addition, hereditary hemochromatosis protein (HFE) is a protein that competes with Tf for binding to the transferrin receptor and modulates the hepcidin level [52]. The latter hormone is increased in most inflammatory-related diseases and infections, thus causing a cellular overload of iron and a low concentration of the metal in the plasma [41].

During infection, there is often active competition between the virus and the host for iron availability, and iron overloading conditions have been reported for hepatitis B/C [53,54] and for HIV [33,55]. HIV-1 patients with thalassemia treated with higher doses of iron chelators have shown longer survival times, suggesting the beneficial role of iron deficiency during viral infection [56,57]. Furthermore, the protective role of iron deficiency has also been reported against malaria [58,59]. Iron overloading can cause redox dyshomeostasis and oxidative stress, resulting in the overactivation of the immune response and immune dysfunction [60,61]. In addition, the dysregulation of iron homeostasis has been described in various pathological states, such as cancer [62,63], auto-immune disease [64,65], and neurodegeneration [66,67], demonstrating that further studies are required to better control the modulation of iron content and to develop novel therapeutic approaches. To date, different therapies have been adopted for the treatment of COVID-19; antiviral treatments and monoclonal antibodies are used for the viral phase of the disease and inhibitors of the inflammatory cascade are used for the inflammatory phase of COVID-19 [68,69,70,71]. However, for COVID-19 and long-COVID conditions, iron-related biomarkers evidencing the molecular and functional mechanisms leading to MIS and ARDS remain to be characterized. Thus, research on predictive biomarkers to identify patients with a high risk of developing ARDS and on prognosis-related biomarkers to follow the evolution of diseases after therapy is a priority.

The aim of this study was to identify candidate proteins that can be used as biomarkers in the study of COVID-19. First of all, we analyzed the molecular changes induced by SARS-CoV-2 in patients’ immune cells with a proteomic approach. This analysis showed a clear alteration of proteins related to iron metabolism. We, therefore, focused the analysis on proteins with a key role in iron metabolism in order to understand the possible molecular mechanisms responsible for redox and iron dyshomeostasis, i.e., Tf, LCN2, HPX, SOD1, Cp, and 5-lipoxygenase (5-LOX). We have previously reported the modulation of 5-LOX with iron cellular content [61], so we analyzed the expression levels of this enzyme and its leukotriene B4 (LTB4) products in relation to COVID-19 and long-COVID patients. Finally, we propose possible biomarkers and molecular mechanisms useful for the therapeutic modulation of redox and iron dyshomeostasis for contrasting MIS in COVID-19 patients. 

## 2. Results

### 2.1. Clinical Characteristics of COVID-19 and Long-COVID Patients

We collected samples from three experimental groups: 25 healthy subjects (control), 30 hospitalized COVID-19 patients (COVID-19), and 10 patients with post-acute infection symptoms (long-COVID). COVID-19 patients had active infection and confirmed diagnosis performed by SARS-CoV-2 real-time reverse transcription–polymerase chain reaction (RT-PCR) with nasopharyngeal swabs. Long-COVID patients showed persistent symptoms such as chronic fatigue, breathlessness, cardiovascular abnormalities, neurocognitive impairments, anxiety, and depression more than 3–6 months after acute infection [15,16,17,18,22,72]. The baseline characteristics and clinical features of COVID-19 and long-COVID patients are reported in Table 1. The gender and age distribution did not differ significantly between the groups (*p* = 1 and *p* = 0.485, respectively). There were no significant differences between the two groups (*p* > 0.05), except for the dyspnea symptom (*p* = 0.002) reported in all long-COVID patients. Some patients showed other chronic morbidities such as hypertension and diabetes mellitus II. Furthermore, hypertension was the most prevalent comorbidity found in COVID-19 and long-COVID patients. Plasma concentrations of interleukin 6 (IL-6) and ferritin were extracted via clinical laboratory testing, and showed no significant differences between the COVID-19 and long-COVID groups (*p* = 0.8683 and *p* = 0.8049, respectively). Iron metabolism is gender specific [73,74], so we analyzed hematological data according to gender in our cohorts of patients. However, we did not find gender-related differences in the blood parameters (Appendix A).

### 2.2. Proteomic and Metabolomic Changes in COVID-19 Lymphocytes

We performed proteomic analysis on lymphocytes CD3^+^ and CD19^+^ isolated from infected patients and healthy controls, as described in the Material and Methods. In total, 221 and 234 proteins were quantified in the infected and healthy pools of CD3^+^ lymphocytes, respectively; likewise, 205 and 161 proteins were identified in the infected and healthy pools of CD19^+^ lymphocytes, respectively. By comparing the protein expression between COVID-19 patients and healthy subjects, we found that 49 and 65 proteins were differentially expressed with statistical significance in CD3^+^ and CD19^+^ lymphocytes, respectively. The proteins involved in iron homeostasis and metabolism for each lymphoid lineage are shown in Figure 1, and were obtained through qualitative networks via the STRING database. In CD3^+^ lymphocytes from COVID-19 patients, protein–protein interactions (PPIs) between HPX, Tf, HFE, and Cp were associated with high confidence (0.7). Those with SOD1 and LTF were associated with medium confidence (0.4), and Erythrocyte Membrane Protein Band 4.2 (EPB42) did not show any interactions. Likewise, in CD19^+^ lymphocytes from COVID-19 patients, PPI interactions with higher confidence (0.7) were found between HPX, Tf, and Cp, while all the other interactions were associated with medium confidence (0.4), except for EPB42 and TTC7A, which did not show any interactions.

### 2.3. Iron-Related Biomarker Proteins Are Dysregulated in COVID-19 and Long-COVID Patients

In order to confirm mass spectrometry protein identification, we performed Western blot analysis on the PBMCs isolated from healthy volunteer donors and COVID-19 patients. Of the proteins identified as differential via proteomic analysis, we selected those that had a major impact on iron and redox homeostasis, such as Tf, LCN2, HPX, SOD1, and Cp. We extended and stratified our data collection to long-COVID patients to further understand the possible role of iron dyshomeostasis during the pathogenesis of COVID-19. Our results showed the dysregulation of all proteins investigated in COVID-19 and long-COVID patients versus healthy controls. Regarding the COVID-19 group, we observed an increasing trend in patients compared to the healthy controls that was found to be significant for Cp, Tf, HPX, and LCN2 (Figure 2A,B). Furthermore, we found that the HPX levels remained significantly elevated in patients with long-COVID compared to the control group. For the other iron-related proteins, we found a decreasing trend for Cp, Tf, and SOD1, and an increasing trend for LCN2 in long-COVID versus controls (Figure 2A,B). Concerning the comparison between COVID-19 and long-COVID patients, generally decreased levels of all iron-related metabolism proteins were reported in long-COVID patients, but only Cp, Tf, and SOD1 were found to be significantly altered.

### 2.4. 5-LOX Expression Is Modulated in COVID-19 and Long-COVID Patients 

The dysregulation of iron-related protein levels reported in Figure 2 is in line with a possible iron overloading condition in COVID-19 patients. We previously reported the essential role of cellular iron in the increase in biological activity of 5-LOX in immune cells [61]. Thus, we investigated if iron dysregulation led to 5-LOX modulation in our patient groups. We measured *5-lox* gene expression and protein levels in COVID-19 patients. As reported in Figure 3A, B, 5-LOX protein levels were downregulated in PBMCs isolated from COVID-19 patients versus healthy controls, and remained at a similar level in long-COVID. In contrast to 5-LOX protein levels, *5-lox* gene expression was found to be upregulated in the nasopharyngeal swabs of COVID-19 patients compared to the healthy controls (Appendix A).

### 2.5. Increased LTB4 and LCN2 Plasma Levels in COVID-19 and Long-COVID-19 Patients

The role of lipid mediators produced by 5-LOX during inflammation has been well characterized in different diseases, but their production in COVID-19 has not yet been investigated. Thus, we measured the systemic levels of LTB4 in order to study the possible post-translational activation of 5-LOX induced by altered iron metabolism. The results showed a significant increase in LTB4 plasma levels in both COVID-19 and long-COVID patients compared to healthy donors (Figure 4A). Furthermore, we extended our analysis to evaluate the plasma levels of LCN2 to investigate if the modulation of this adipokine at the systemic level reflected the increase at the cellular level reported here (Figure 2). As shown in Figure 4B, we found that LCN2 plasma levels were significantly increased in COVID-19 patients compared with long-COVID patients and healthy donors. Long-COVID patients had similar levels of this adipokine with respect to the healthy controls, according to the cellular Western blot data (Figure 2). The achieved Power value was 0.8, and was calculated using the G*Power application (version 3.1) (The G*Power Team, Heinrich Heine University, Düsseldorf, Germany) [75].

## 3. Discussion

Dysregulated levels of iron-related biomarkers are associated with various pathological conditions [62,76], including pulmonary and respiratory disease [77,78], and are also reported in hyperferritinemic syndromes (HFS) [79,80]. HFS comprise a spectrum of diseases characterized by high levels of serum ferritin (ferritin > 300 ng/mL) [81], such as macrophage activation syndrome (MAS), Gaucher disease, adult-onset Still’s disease (AOSD), rheumatoid arthritis, and catastrophic antiphospholipid syndrome (CAPS) [80,82,83]. Hyperferritinemia has been shown to be relevant for viral diseases [33,84] and predicts poor outcomes in patients affected by influenza A [85] and Crimean–Congo hemorrhagic fever [86]. The cytokine profiles and the extremely high levels of ferritin reported in HFS are in line with the results reported here for COVID-19 patients (Table 1), confirming that COVID-19 is an HFS [87,88,89]. During viral infection, two principal mechanisms occur to modulate ferritin levels, iron availability, and inflammatory cytokines levels: IL-1β and IL-6 [39,84]. Elevated ferritin levels can persist several months after the onset of COVID-19 in some cases [90,91], as also reported in our cohort of long-COVID patients (Table 1). Raised serum ferritin levels can induce hepatic cell death, triggering an increase in free iron systemic levels after iron release from ferritin [92,93]. Notably, in hospitalized COVID-19 patients, high ferritin levels and iron dysregulation were associated with an adverse clinical course [64,91,94]. To date, it is not clear if disturbances of iron handling are just an adaptation response to SARS-CoV-2 infection or are involved in the pathophysiological mechanism, and further studies are required [95,96]. In the immune response to COVID-19, ferritin levels increase dramatically leading to the condition of hyperferritinemia [87,88,89]. This iron-based resistance mechanism to pathogens involves a systemic reduction in circulating iron due to the inhibition of cellular iron export and the induction of cellular iron import [33,41]. In particular, this mechanism is activated by the synthesis and secretion of hepcidin by hepatocytes, which are in turn influenced by iron levels in the body as well as inflammation, high levels of HFE, erythropoiesis, and hypoxia [38,41]. Thus, under these inflammatory conditions, the hepatic hormone hepcidin is upregulated, reducing cellular iron efflux into the blood lumen by binding to FPN [37,39,41]. The blockage of FPN decreases iron absorption from the intestine and results in iron being retained in the macrophages, which might explain the low serum iron concentration and hypoferremia as a reaction to the infection [33,97]. On the other hand, an increase in intracellular iron can lead to an increase in inflammation via the iron-dependent intracellular post-translational activation of 5-LOX and its LTB4 products (Figure 4A) without any increase in protein levels (Figure 3). In our study, we individuated iron-related metabolism proteins differentially expressed in COVID-19 and long-COVID patients in comparison to healthy donors (Figure 1 and Figure 2). We found that the amounts of Cp, Tf, HPX, LCN2, and SOD1 increased in PBMCs isolated from COVID-19 patients, compared to both the long-COVID group and the control group (Figure 2). The functional interplay between Tf and Cp has an important role in intestinal iron absorption and in iron systemic transport; the bilobate Tf can bind to ferric iron (Fe^3+^) and represents the major transporter of the metal in plasma, while Cp modulates the loading of iron into Tf by catalyzing the oxidation of the ferrous form (Fe^2+^) into Fe^3+^ [40,98,99,100]. Our data reported increased levels of both proteins in COVID-19 patients (Figure 2), suggesting the activation of a possible protective mechanism against the iron overload reported during SARS-CoV2 infection [92,95,101] by binding Fe^3+^ and oxidating Fe^2+^ to less toxic ferric forms [40,102]. Excessive Fe^2+^ catalyzes the Fenton and Haber–Weiss reactions and promotes the accumulation of reactive oxygen species (ROS) and, in particular, the formation of hydroxyl radical (⋅OH) [103,104]. This condition promotes oxidative stress, and can also be responsible for hemolysis, a common complication reported in COVID-19 cases [105,106,107]. During the acute phase of SARS-CoV-2 infection, and in some cases after the resolution of infection, a spectrum of hematological/hemolytic complications has been reported, characterized by the increased destruction of red blood cells (RBCs) [108,109]. The hemolytic process induces an increase in the free hemoglobin, heme, and Fe^2+^ described in COVID-19 patients, and concurs with increasing inflammation and oxidative stress [108,109]. Our results reported increased HPX levels in COVID-19 and long-COVID patients (Figure 2); this protein is known to protect cells from free heme toxicity occurring during hemolysis [48,49]. Indeed, in animal models of ARDS and chronic pulmonary disease, the administration of HPX attenuates inflammation and lung fibrosis [110,111]. We speculate that the increased HPX levels reported here could act protectively by binding not only free heme, but also the RBC membrane, preventing further hemolysis [48,110]. Furthermore, the induction of ROS formation could also be responsible for enhanced SOD1 levels (Figure 2), one of the most powerful antioxidant enzymes in the first line against ROS [51,112]. Previous works have reported an increase in antioxidant enzyme activities, such as SOD and catalase, during SARS-CoV2 infection [113,114], and animal studies have shown that the use of a synthetic and stable SOD has a protective effect in pulmonary fibrosis, lung inflammation, and ARDS [115,116,117]. However, the redox status and complete profiling of oxidative stress markers in COVID-19 and in long-COVID patients are not yet described. Some studies reported no changes between the serum activities of SOD and CAT enzymes in COVID-19-infected patients [118], or an inhibition of antioxidant systems, leading to a decrease in the overall antioxidant capacity [119,120]. In addition, the ingenuity pathway analysis (IPA) performed with our proteomic data highlighted the upregulation of oxidative stress-related biological functions such as the “metabolism of hydrogen peroxide” in COVID-19 patients versus healthy controls, driven by the pool of proteins as reported in Appendix A.

Oxidative stress pathways could also be responsible for somatic and mental symptoms of long-COVID. Depression, generalized anxiety disorder, and chronic fatigue are manifestations of oxidative stress-activated pathways and are coupled with lowered antioxidant defenses [121,122,123]. Previous studies have suggested connections between redox imbalance/oxidative damage and long-COVID symptoms caused by a reduction in antioxidant defense mechanisms [124,125]. In line with this theory, in our cohort of long-COVID patients, we found a decreasing trend of antioxidant proteins, such as Cp, Tf, and SOD1 (Figure 2) [42,51,126], in comparison to COVID-19 patients and healthy controls, and a significant increase in HPX levels compared to the healthy controls (Figure 2). These results suggest that oxidative stress is a pivotal point in the pathophysiology of COVID-19, as well as in long-COVID. Indeed, in patients affected by long-COVID, the imbalance between the low concentration/activity of antioxidant proteins Cp, Tf, and SOD1 and the increased hemolytic crisis, suggested by the upregulated HPX levels (Figure 2), could be responsible for increased levels of ROS and the aberrant oxidative stress reported after recovery from SARS-CoV2 infection [127,128]. 

Our data show a possible connection between the redox imbalance reported in COVID-19 and long-COVID and altered iron metabolism. Interestingly, it has been reported that susceptibility to viral infection with HIV, H1N1, SARS, and COVID-19 is associated with iron levels [129,130,131], and increased plasma levels of free iron correlate with adverse outcomes for COVID-19 patients [132,133]. Previous studies have demonstrated the anti-viral effects of iron-chelators, such as deferoxamine (DFO) or deferiprone, for HIV, HSV-1, and CMV [56,57,134,135], and iron chelation therapies are shown to be effective in the management of COVID-19 patients by decreasing the production of free radicals and reducing IL-6 levels [129,136]. Furthermore, iron excess has been reported to be involved in the increased lipoxygenase activity described in immune cells during inflammation [61] and in the cell death mechanism triggered by iron-catalyzed lipid peroxidation, known as ferroptosis [61,79]. Here, we report a significant increase in the gene expression levels of 5-LOX in COVID-19 patients (Appendix A), in line with previously published data from other groups [137], and a downregulation of protein amounts in COVID-19 and long-COVID patients in comparison to healthy donors (Figure 3A,B). Concerning the 5-LOX activation state, we report a significant increase in LTB4 plasma levels in COVID-19 and long-COVID patients versus healthy donors (Figure 4A), which seems to be in line with the previously described post-transcriptional activation of an apo-form of 5-LOX, which leads to an active holo-5-LOX able to produce LTB4 in iron overloading conditions [61]. It is fundamental to underline that 5-LOX expression is complex and its modulation is associated with the cytokine profile. The important link between IL-4 and LOXs during the inflammatory/immune response has been well described [138,139]. Furthermore, Spanbroek and colleagues described a cytokine-specific modulation of 5-LOX, reporting that prolonged stimulation with IL-4 downregulates 5-LOX protein levels in dendritic cells and other leukocytes [140]. Thus, we speculate that the persistent high levels of IL-4 reported during SARS-CoV-2 infection and in the post-acute phase [29,141] are responsible for the downregulation of the 5-LOX protein in COVID-19 and long-COVID patients, compared with healthy donors (Figure 3A,B). 

LTB4 is one of the most important candidates responsible for the hyperimmune/inflammatory response in the progression of COVID-19 due to its chemoattractant properties and capability to carry lymphocytes out to airways [142,143,144]. During acute COVID-19 infection, LTB4 could act protectively by suppressing viral replication [145,146] and inducing leukocyte recruitment. This occurs in other viral infections, including the herpes virus, CMV, and influenza [145,147]. However, it has been reported that aberrant and chronic LTB4 production can induce an uncontrolled release of chemokines and cytokines, causing blood lymphocytopenia, as described in COVID-19, and could be detrimental to host defense [148,149]. In different chronic inflammatory diseases, including autoimmune diseases, allergy, obesity, and chronic infection, excessive plasma LTB4 levels could propagate pathological inflammation in affected tissues, thus contributing to tissue injury [148,150,151,152]. In long-COVID patients, a low and continuous grade of inflammation has been reported [153,154], mimicking the conditions occurring in chronic disease [155]. In line with these data are the elevated LTB4 plasma levels reported here in relation to long-COVID patients (Figure 4A). Furthermore, LTB4 systemic levels were found to be associated with the severity grade of COVID-19 in patients with diabetes [137], and increased LTB4 production has been reported in immune cells after SARS-CoV2 infection [156]. 

LCN2 is a multifaceted protein member of the adipocytokines with a well-characterized bacteriostatic role [157,158], and its association with viral infections has been described [159,160]. LCN2, also known as neutrophil gelatinase-associated lipocalin (NGAL) or siderocalin [47,161], is upregulated in several immune disorders [162,163,164]. It is measurable in biological fluids during viral infection and inflammation states [45,158,165,166], and its role as an iron regulatory protein has recently emerged [45,158,167]. LCN2 protects cells against oxidative stress [50], and during iron overload conditions, its expression is upregulated both at the cellular and systemic level [168,169]. The defensive role of LCN2 against iron excess and oxidative stress is related to its ability to indirectly bind iron [170,171,172], to induce the expression of antioxidant molecules (including SOD1 [173,174]), and its intrinsic antioxidant properties [169]. In our work, we measured an increase in circulating plasma LCN2 levels in the cohort of COVID-19 versus long-COVID patients and healthy controls (Figure 4B), parallel to the data reported here concerning the protein amounts in PBMCs (Figure 2). Given the observation that the cellular and systemic LCN2 forms were significantly elevated in COVID-19 patients, we propose that during SARS-CoV2 infection, an induction of LCN2-mediated protection against iron-induced toxicity can occur.

Finally, SARS-CoV-2 infection induces the well-described cytokine storm responsible for, among other things, high ferritin levels and mitochondrial dysfunction, leading to oxidative stress [87,88,89]. These events contribute to modulating iron levels and induce significant changes in the proteins responsible for iron metabolism. Overall, our findings suggest that iron dyshomeostasis causes an increase in ROS levels, oxidative stress, and the hemolytic process, which in turn can increase free iron and heme levels, as well as cellular iron overloading and the post-translational activation of 5-LOX (see Figure 5). Here, we speculate on the presence of the following defensive mechanisms that could occur against free iron/heme toxicity and oxidative stress during COVID-19: (i) Tf, Cp, and LCN2 are increased to protect against free iron by reducing Fe^3+^; (ii) the antioxidant enzyme SOD1 is upregulated to reduce ROS levels and modulate LCN2 levels; (iii) HPX protein levels are enhanced to protect against free heme toxicity and prevent further hemolysis (Figure 5).

These data suggest further investigations are needed to evaluate the role of different iron-related proteins and 5-LOX activation in an increased number of COVID-19 and long-COVID patients in order to better assess the correlation between disease progression and severity. Indeed, a limitation of this study is the number of subjects included in the two cohorts of patients, particularly for long-COVID. The small size cohort for long-COVID analyzed in this study determines the exploratory and preliminary nature of our results. Further analyses are required in a larger cohort of patients in order to better validate LTB4 and LCN2 as a potential biomarker for COVID-19 and long-COVID. In conclusion, these data strongly suggest the need to extend the clinical analyses of COVID-19 patients in the context of the iron-related proteins reported here for a better evaluation of the inflammatory state and disease progression of the patients, and to develop innovative therapeutical approaches.

## 4. Materials and Methods

### 4.1. Patients and Sample Information

Samples were collected from the Center for Advanced Studies and Technology (CAST), “G. d’Annunzio” University of Chieti-Pescara from patients hospitalized in the Infectious Disease and Pneumology Unit of the S.S. Annunziata Hospital of Chieti-Pescara with COVID-19 and long-COVID and from healthy volunteer donors. All subjects gave their informed consent for inclusion before they participated in the study. The study was conducted in accordance with the Declaration of Helsinki, and the protocol was approved on 05 May 2022 by the Ethics Committee of “G. d’Annunzio” University. Electronic data regarding epidemiological, demographic, and clinical symptom laboratory tests were extracted and are reported in Table 1. Regarding COVID-19 infection, SARS-CoV-2 was confirmed with nasal and pharyngeal swab specimens using RT-PCR assay, as described below. All samples of blood were obtained from patients before they received any therapy. The material for analysis was collected from 30 hospitalized COVID-19 patients (COVID-19 group) and from 10 patients with persistent symptoms post-acute infection (long-COVID-19 group). Patients were included in the COVID-19 group if they had a diagnosis of SARS-CoV-2 infection, and showed at least one of the common symptoms at the onset of illness such as fever, cough, or dyspnea [5,6,7,8], and after thorough evaluation and discussion among an interdisciplinary clinical team. Regarding COVID-19 patients, infection with SARS-CoV-2 was confirmed via nasopharyngeal swab specimens using an RT-PCR assay, as described hereafter. To characterize the post-acute infection phase, we enrolled patients with persistent symptoms such as chronic fatigue, breathlessness, cardiovascular abnormalities, neurocognitive impairments, anxiety, and depression more than 3–6 months after acute infection in the long-COVID group [15,16,17,18,22,72]. In particular, long-COVID patients had recovered from mild-to-severe COVID-19 illness and reported a mixture of the previously described symptoms that could not be explained by alternative diagnoses [72,175]. In addition, 25 healthy volunteers were enrolled (control group). To be included in the study, healthy donors were required to be free of clinically significant diseases or medical conditions. For all groups, patients were excluded if they had hemodynamic instability, acute severe multiorgan failure, or an expected survival of less than 3 days.

### 4.2. Peripheral Immune Cell (PBMC) Isolation by Fluorescence-Activated Cell Sorting 

Peripheral blood samples, collected using ethylene diamine tetra acetic acid (EDTA) as an anticoagulant, were stained for flow cytometry purposes within biosafety level 2 laboratories. Each sample underwent a fixation/erythrocyte lysis step using Lysing solution (Becton Dickinson, BD, Biosciences, La Jolla, CA, USA) under gentle agitation (15 min, room temperature). Samples were then stained by adding the reagent mix reported in Appendix A (30 min, 4 °C), washed, and postfixed (BD Biosciences, La Jolla, CA, USA); then, CD3^+^ T lymphocytes and CD19^+^ B cells were separated via fluorescence-activated cell sorting (FACSAria III, BD Biosciences, San Jose, CA, USA) using a 100 μm nozzle [176]. A high level of purity for each isolated population was achieved (>90%).

### 4.3. Proteomics Analysis and Data Processing

After cell sorting, an increase in the protein concentration of samples was achieved by loading CD3^+^T and CD19^+^B lymphocytes onto Amicon^®^ Ultra Centrifugal filters (Merck Millipore, Darmstadt, Germany), followed by centrifugation at 4000× *g* rpm for 1 h. Cells were resuspended in a lysis buffer containing 6 M urea, 100 mM tris base, 1% Triton X-100, 50 mM dithiothreitol (DTT) and 0.25% 3-[(3-cholamidopropyl)dimethylammonio]-1-propanesulfonate (CHAPS). Residuals from cell debris and insoluble protein particulates were removed by centrifugation at 4000× *g* rpm for 5 min. Cells were disrupted by pulse sonication on ice with an amplitude of 50%. An infected pool derived from three adult patients with COVID-19 and a healthy pool from two adult donors with negative test results by RT-PCR and IgG serological analysis, each containing 100,000 sorted cells, were obtained for both CD3^+^T and CD19^+^B lymphocytes. A filter-aided sample preparation (FASP) digestion protocol was performed using 50 mM iodoacetamide as an alkylating agent, and trypsin was added to a final substrate-to-enzyme ratio of 50:1 (*v*/*v*).

Then, 4 μL of extracted peptide from each sample was analyzed in triplicate via liquid chromatography tandem mass spectrometry (LC–MS/MS) using a Dionex UltiMate 3000 RSLCnano System (Thermo Fisher Scientific, Waltham, MA, USA) coupled to an Orbitrap Fusion Tribrid mass spectrometer (Thermo Fisher Scientific, Waltham, MA, USA). Peptides were loaded on the PepMap 100 C18 trap cartridge (300 μm I.D., 5 mm L., 5 μm ps, Thermo Fisher Scientific, Waltham, MA, USA) and subsequently separated on an EASY Spray C18 (75 μm I.D., 250 mm L., 2 μm ps, Thermo Fisher Scientific, Waltham, MA, USA) analytical column. The flow rate was set to 300 nL/min with a total run time of 65 min and the following chromatographic gradient: from 5 to 25% of B for 40 min followed by 25 to 55% for 5 min; from 55 to 90% for 1 min and maintenance at 90% B for 2 min; from 90 to 5% B in 0.5 min and maintenance at 5% B for 1 min; and the repetition of ramping steps 90–5% B twice to wash the column until the end of the run. Mobile phase A was 0.1% formic acid (FA) in water (H_2_O) and mobile phase B was 0.1% formic acid in acetonitrile (ACN). The mass spectrometer (MS), operating with positive ion polarity and data-dependent acquisition (DDA), was equipped with a nanoESI spray source. Precursors in the range 375 to 1500 m/z with a preferred charge state +2 to +5 and absolute intensity above 1.0 × 10^4^ were selected for higher energy collision dissociation (HCD) fragmentation. Proteomics raw data were processed as previously described [177].

A false discovery rate (FDR) of 1% was applied both at the protein and peptide levels. The retention times of all analyzed samples were linearized with the “Match between runs” algorithm of MaxQuant, which boosts the number of identifications for peptides that are present in different samples but not uniformly identified via MS/MS, with a retention window of 0.7 min and an alignment time window of 20 min. Intensity-based absolute quantification (IBAQ) was used to quantify protein abundance in each sample. Statistical analysis was performed with Perseus version 1.6.15.0 (Max-Planck Institute for Biochemistry, Martinsried, Germany). IBAQs were log2-transformed to facilitate the calculation of the protein expression. The minimum number of valid values accepted was set at 2 in at least one group in order to simultaneously evaluate the differential proteins and the presence and absence of proteins between different conditions. The univariate statistical analysis was conducted with a *p*-value threshold of 0.05.

The STRING database was used to highlight the physical and functional interactions between the identified proteins in each condition.

### 4.4. Western Blot Analysis

PBMCs were lysed for Western blot analysis, as previously described [178]. Cellular protein amount was determined using the Bradford assay (Bio-Rad, Hercules, CA, USA); cell lysates were denatured in 5X Laemmli sample buffer at 98 °C for 10 min and equal amounts of proteins were separated using 12, 10, or 8% SDS-PAGE and transferred to a PVDF membrane (Merck Millipore, Darmstadt, Germany). The membranes were blocked in 5% not-fat dry milk in PBS with 0.01% Tween 20 for 1 h at room temperature, and then incubated overnight at 4 °C with the following antibodies: anti-5-LOX (#3289, Cell signaling, Danvers, MA, USA), β-actin (#37100, Cell signaling, Danvers, MA, USA), anti-CP (#98971, Cell signaling, Danvers, MA, USA), anti-Tf (#82411, Abcam, Cambridge, UK), anti-LCN2 (#44058, Cell signaling, Danvers, MA, USA), anti-SOD1 (#2770, Cell signaling, Danvers, MA, USA), and anti-HPX (#124935, Abcam, Cambridge, UK). Subsequently, the membranes were incubated for 1 h at room temperature with HRP-conjugated secondary antibodies, and were detected with an enhanced chemiluminescence (ECL) solution (Pierce, Thermo Fisher Scientific, Waltham, MA, USA).

### 4.5. Quantitative Real-Time Reverse Transcription–Polymerase Chain Reaction (qRT-PCR) Analysis 

Nasopharyngeal samples were used for RNA extraction using the MagMAX Viral/Pathogen II (MVP II) nucleic acid isolation kit (Thermo Fisher Scientific, Waltham, MA, USA), and an automated KingFisher magnetic particle processor (Thermo Fisher Scientific, Waltham, MA, USA), as indicated in the manufacturer’s instructions. Extracted RNA underwent real-time reverse transcription–polymerase chain reaction (qRT-PCR) using the TaqPath™ COVID-19 CE-IVD RT-PCR kit assay (Thermo Fisher Scientific, Waltham, MA, USA) according to the manufacturers’ protocols. The QuantStudio 5 Real-Time PCR System assay (DX) was used for qRT-PCR analysis (Thermo Fisher Scientific, Waltham, MA, USA) and was used to analyze three different viral gene targets: *ORF1ab*, *N*, and *S* genes. A specimen was considered positive in the presence of amplification of at least two of the three target genes. Extracted RNA was also used to identify 5-LOX gene expression, followed by cDNA synthesis using the RT^2^ First Strand kit (QIAGEN, Hilden, Germany according to the manufacturers’ protocols. The obtained cDNA was taken for real-time PCR using SsoAdvanced Universal Taqman Supermix (Biorad, Hercules, CA, USA ) and Bio-Rad PrimePCR primers on CFX Real-Time PCR Detection Systems (Biorad, Hercules, CA, USA ). The following PCR program was used: 95 °C for 30 s; 40 cycles of 95 °C for 15 s and 60 °C for 30 s. All gene expressions were normalized using human glyceraldehyde-3-phosphate dehydrogenase (GAPDH) as a reference gene. Differences in threshold cycle (Ct) number were used to quantify the relative amount of PCR target genes. Relative amounts of different gene transcripts were calculated using the ΔΔCt method and were converted to the relative transcription ratio (2^−ΔΔCt^) for statistical analysis.

### 4.6. Enzyme-Linked Immunosorbent Assays 

The LTB4 and LCN2 content in the plasma from COVID-19, long-COVID patients, and healthy controls was determined with enzyme-linked immunosorbent assays (ELISA) according to the provided instructions. ELISA kits for LTB4 and LCN2 were purchased from Abcam (#ab133040 and #ab113326, Cambridge, UK). Briefly, for LTB4 well plates, LTB4 AP conjugate and antibody anti-LTB4 were incubated for 2 h at room temperature. After three washes, 200 μL of the pNpp substrate solution was added for 2 h at 37 °C. Then, the reaction was stopped and read at 405 nm with the Elisa plate reader. Briefly, for LCN2, well plates were incubated with the sample and standard overnight at 4 °C. After three washes, biotinylated detection antibody anti-LCN2 was added and incubated for 1 h at RT. After washing, 100 μL of HRP-Streptavidin solution was added for a total time of 45 min. Substrate-stabilized reagent was added for at least 30 min in the dark, before the reaction was stopped with the addition of 1N H2SO4. The resulting color was read at 450 nm with the Elisa plate reader. 

### 4.7. Statistical Analysis

Statistical analysis was performed using GraphPad Prism 9 software (GraphPad Software, San Diego, CA, USA). The Student *t*-test was adopted for parameters with normal distribution, the Mann–Whitney test was adopted for non-parametrical distribution, and the chi-squared test for categorical parameters was used to compare groups, as reported in the figure legends. Calculated *p* values of less than 0.05 were considered significant. Data are reported as the mean of three or two independent experiments ± SEM, as indicated in the figure legends. 

## Figures and Tables

**Figure 1 ijms-24-00015-f001:**
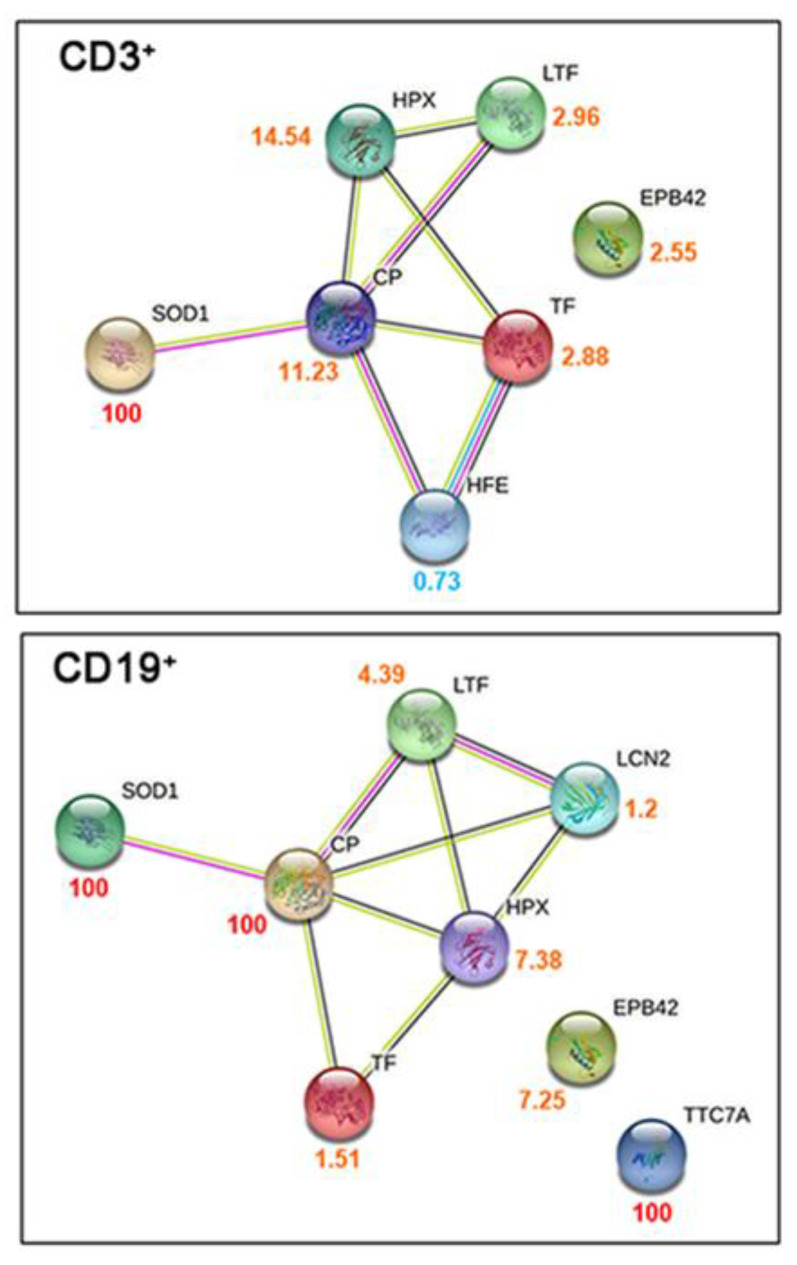
Characterization of proteome lymphocyte measurements taken from COVID-19 patients. Qualitative STRING networks from gene ontology analysis of the iron-related metabolism proteins quantified in CD3^+^ and CD19^+^ sorted lymphocytes. The numbers beside each protein represent the ratio between the protein abundance in COVID-19 patients and controls and are colored accordingly (red for proteins quantified only in COVID-19 patients, orange for proteins with higher abundance in COVID-19 patients, and light blue for proteins with higher abundance in controls).

**Figure 2 ijms-24-00015-f002:**
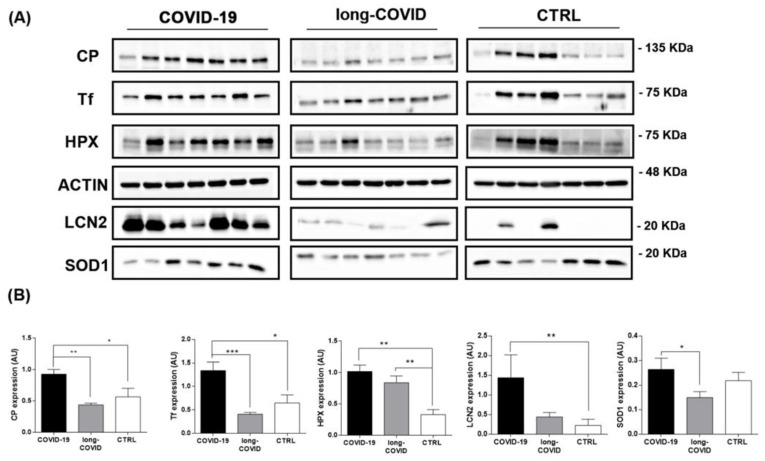
Western blot analysis for iron-related proteins in PBMCs isolated from COVID-19, long-COVID patients, and healthy controls. (**A**) Panels report the proteins levels for CP, Tf, HPX, LCN2, SOD1 and ACTIN in PBMCs isolated from COVID-19 patients (n = 7), long-COVID (n = 7) patients, and healthy volunteer donors (n = 7). (**B**) Band intensities were quantified using Image J and are reported in the figure as arbitrary units (AU). Data reported in this figure are the mean ± SE of two independent experiments (* *p* < 0.1, ** *p* < 0.01, *** *p* < 0.001).

**Figure 3 ijms-24-00015-f003:**
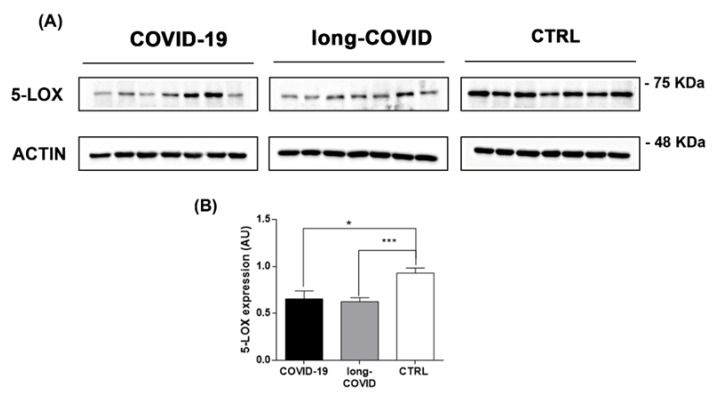
5-LOX is modulated in COVID-19 and long-COVID patients. (**A**) 5-LOX protein levels were analyzed in PBMCs isolated from COVID-19 patients (n = 7), long-COVID (n = 7), and healthy donors (n = 7) using Western blot, and (**B**) bands intensities were quantified using Image J and reported as arbitrary units (AU). Data reported in this figure are the mean ± SE of two independent experiments (* *p* < 0.1, *** *p* < 0.001).

**Figure 4 ijms-24-00015-f004:**
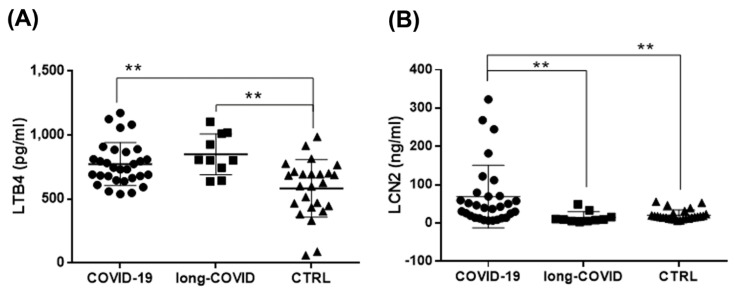
Increased LTB4 and LCN2 systemic levels in COVID-19 and long-COVID patients. (**A**,**B**) Plasma levels of LTB4 and LCN2 in COVID-19 patients (n = 30), long-COVID-19 patients (n = 10), and healthy donors (n = 25) were analyzed using ELISA. Data reported in this figure are the mean ± SE of two independent experiments (** *p* < 0.01).

**Figure 5 ijms-24-00015-f005:**
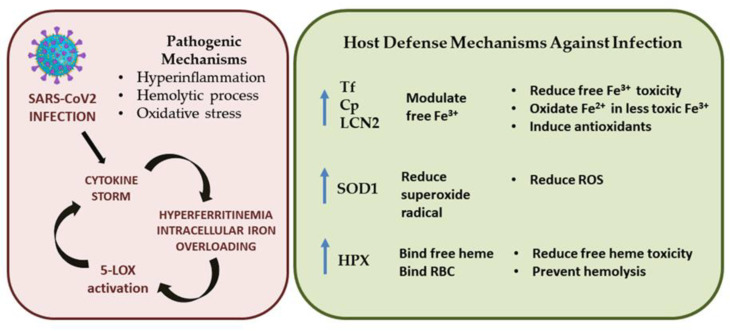
Schematic representation of pathogenesis of SARS-CoV-2 infection and possible defense mechanisms.

**Table 1 ijms-24-00015-t001:** COVID-19 (n = 30) and long-COVID (n = 10) patient characteristics.

Variable	Value in COVID-19 (n = 30)	Value in Long-COVID (n = 10)	*p*-Value
**Gender (Female), n (%)**	14 (46.7%)	5 (50%)	n.s.
**Age, mean ± SD**	67.5 ± 14.8	62.7 ± 13.2	n.s
**Symptoms, n (%)**			
Fever	20 (66.7%)	3 (30%)	n.s.
Cough	8 (26.7%)	3 (30%)	n.s.
Dyspnea	13 (43.3%)	10 (100%)	0.002 **
Diarrhea	2 (6.7%)	-	n.a.
Asthenia	4 (13.3%)	-	n.a.
**Comorbid Conditions, n (%)**			
Any	-	3 (30%)	n.a.
Hypertension	15 (50%)	3 (30%)	n.s.
Diabetes Mellitus II	5 (16.7%)	2 (20%)	n.s.
Hypothyroidism	1 (3.3%)	1 (10%)	n.s.
Cancer	5 (16.7%)	-	n.a.
Chronic kidney disease	3 (10%)	-	n.a.
Obesity	2 (6.7%)	-	n.a.
**IL-6 (pg/mL)**			
Median value (min.–max.)	128.6 (3.6–278.9)	64.8 (11.1–194)	n.a.
Cut-off < 6.4 pg/mL			
**Ferritin (ng/mL)**			
Median value (min.–max.)	991.8 (37.2–4265)	1608.5 (205–4638)	n.a.
Cut-off 22–274 ng/mL			

Demographic and clinical characteristics of COVID-19 and long-COVID patients included in the study. Results were obtained using the Mann–Whitney test for non-parametrical distribution and the chi-squared test for categorical parameters. Values are expressed as mean ± SD or median (min-max). Cut-off values are given for an appropriate laboratory. ** *p* < 0.01; n.a.—not available; n.s.—not significant.

## Data Availability

The data presented in this study are available in this article.

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
