# Peer review of "Iron Dyshomeostasis in COVID-19: Biomarkers Reveal a Functional Link to 5-Lipoxygenase Activation"

_ijms, 2022, doi:10.3390/ijms24010015_

Round 1

Reviewer 1 Report

will be sent separately.

Author Response

Responce to Reviewer 1 Comments

Dear Reviewer,

We thank you for your constructive criticisms and suggestions, which allowed us to significantly improve the manuscript. We hope the manuscript can be accepted in this revised form.

Changes responding to all the Reviewers are yellow highlighted in the revised text.

Please find here following our point-by-point answers.

Point 1.

Proteomic analysis of lymphocytes with the proteins ceruloplasmin (Cp), transferrin (Tf), hemopexin (HPX), lipocalin 2 (LCN2) and superoxide dismutase 1 (SOD1), 5- lipoxygenase (5-LOX), leukotriene B4 (LTB4) and lipocalin 2 (LCN2) in the focus as possible markers for COVID-19 and long Covid. All these markers are related to serum irom or serum iron metabolism. Immune modulators commonly discussed at this time are excluded (see e.g. Distinguishing features of long Covid identified through immune profiling by Jon Klein et al. medRxiv preprin, with decreased levels of cortisol being the most significant individual predictor).

Response 1:

We thank the Reviewer for his/her comment, and we consequently improved the Introduction section (see revised manuscript: lines 69-77, page 2).

Point 2:

The authors introduce their focus on the biomarkers transferrin (Tf), lipocalin 2 (LCN2), hemopexin (HPX), superoxide dismutase 1 (SOD1), ceruloplasmin (Cp) and 5-lipoxygenase (5-LOX) and leukotriene B4 (LTB4). All are related to body iron. At this point it would be very helpful to give a short introduction into the functioning of the biomarkers considered and their relation to body iron.

Response 2:

We thank the Reviewer for his/her comment, and we consequently improved the Introduction section (lines 81-101, page 3).

Point 3:

I am astonished to learn that about 80% of the Covid patients develop long Covid, that means that almost all Covid-19 patients develop long Covid symptoms

Response 3:

We thank the Reviewer for the opportunity to better clarify this point. There is a lack of harmonization in terminology and data regarding the patients affected by long-COVID. To further underline this concept, we rephrased the sentences in Introduction section (lines 62-65, page 2).

Point 4:

2.1 Clinical characteristics of COVID-19 and long-COVID patients 25 controls, 30 Covid, and 10 long-Covid patients. Table 1 for an easy comparison values of the control group should be included.

Response 4:

We thank the Reviewer for his/her comments. We have now improved this part better describing the clinical characteristics, moreover we added inclusion/exclusion criteria for the control group in Material and Methods section (lines 472-483, 485-487, page 14). We don’t have plasma concentrations of interleukin 6 and ferritin for the control group since this test were performed only for patients in the clinical service.

Point 5:

The ferritin values for both Covid and long Covid are exteremely high and do not show a tendency (see Serum Metabolic Profile in Patients With Long-COVID (PASC) Syndrome: Clinical Implications by Evasio Pasini et al. Frontiers in Medicine 8 (2021)). Since the iron status in males and females is different, this difference should be considered.

Response 5:

We thank the Reviewer for the opportunity to better describe the hematological parameters of COVID-19 and long-COVID patients. We added Supplementary Figure 1, which describes the data analysis according to gender, and we implemented paragraph 2.1 Clinical characteristics of COVID-19 and long-COVID patients, in Results section (lines 151-154, page 4).

Point 6:

2.2. Proteomic and metabolomic changes in COVID-19 lymphocytes

The String presentation in Fig. 1 should be explained in more detail or omitted

Response 6:

We thank the Reviewer for the opportunity to better describe the Figure 1 (line 174-182, page 5) and Figure 1 legend (line 191-194, page 6) in 2.2. Proteomic and metabolomic changes in COVID-19 lymphocytes, in Results section.

Point 7:

2.3. Dysregulated iron metabolism in COVID-19 and long-COVID patients

Are the results presented a dysregulation of iron metabolism or present the authors changes in the body iron related marker molecules?

Response 7:

We agree with the point raised by the Reviewer. Consequently, we decided to change the title for 2.3 paragraph in Results section in “iron-related biomarker proteins are dysregulated in COVID-19 and long-COVID patients.” (line 196, page 6).

Point 8:

 The authors present a long and extensive discussion, however, I miss the discussion of the hepcidin- ferroportin axis in health and disease, the ferritin storm or hyperferritinemia and the hypoferremia.

Under inflammatory conditions hepcidin is upregulated and binds to ferroportin, closing the iron gates so that the intracellular iron pool is withheld from export into the blood lumen.The blockage of ferroportin decreases the iron absorption from the intestine and leads to a retention of iron in macrophages which might explain the low serum iron concentration, hypoferremia, as reaction of the infection. Low levels of serum iron should lead to a decrease in virus proliferation during the infection period. In the immune response period the ferritin level increases dramatically, ferritin storm or hyperferritinemia. The fate and metabolism of this ferritin is so far not really understood. See Theresa Hippchen et al. Hypoferremia is Associated With Increased Hospitalization and Oxygen Demand in COVID-19 Patients. HemaSphere (2020) 4:6(e492). http://dx.doi.org/10.1097, and others.

Fig. 5 which acts like a conclusion: the cytokine storm with the subsequent hyperferritinemia is OK. But I thought there is also a hypoferremia and not an iron overload! For me that is the problem: during the immune reaction phase serum iron is needed, but cannot be given orally because the ferroportin export is not available as long as hepcidin is upregulated. Even, if all ferritin iron is transferred to serum iron it would not be enough. We have to wait for further studies looking at these processes.

Response 8:

We agree with the Reviewer implemented this part in the discussion (lines 289-303, page 10) and modified the Figure 5.

Thus, in the revised manuscript we described the hepcidin-ferroportin axis in health and disease, hyperferritinemia and the hypoferremia into the discussion section (lines 289-303, page 10). This allowed us to better describe the iron dyshomeostasis condition responsible for the increase of the intracellular iron required also for the activation of 5-LOX. However, as we concluded in our manuscript, we agree with the Reviewer that further studies are required to better understand the role of the different iron-related proteins and 5-LOX activation in order to better assess the correlations with disease progressions and severity.

Reviewer 2 Report

The topic of the research article is of great interest. However, I would not recommend publishing the article in its current format as it requires lots of improvement. The main drawbacks of this manuscript Below are several specific comments:

1.
The whole manuscript is mixed between American English and British English, the authors even use British English or American English. For consistency, consider replacing it with the American English spelling.

2. The English writing should be further improved, as there are many grammatical or typing errors. It is suggested to ask a native speaker to polish it.

3. The bibliographic references should be revised according to the journal's instructions

4. Minor points

Line 26 changes The aim of this study is to This study aims

line 30 changes analysis to analyses

line 31 adds a comma before and

Line 35 changes suggesting to suggest

line 48 add a comma before or

line 53 changes diarrhoea, to diarrhea, and haemoptysis to hemoptysis

line 53 adds a comma before and

line 54 add a comma before and

line 63 adds the before definition

line 63 adds a or the before long-COVID

line 66 adds a comma before and

line 66 adds for before months

line 71 removes an before active

line 74 changes thalassaemia to thalassemia

Line 77 removes a before redox

line 78 changes an before overactivation to the and removes in before immune

line 79 removes a before  dysregulation

line 80 adds a comma before and

line 82 adds a comma after date

line 86 changes iron related to iron-related

line 87 removes still before remain

line 88 adds a before high

Line 91 changes analysed to analyzed

line 93 adds a comma before and after therefore

line 95 removes: after as

Line 96 adds a comma before and

line 97 removes a before modulation

line 107 removes the before age

Line 129 changes was to were

Line 141 changes coloured to colored

Line 147 add a comma before we

line 148 removes : after as and adds a comma before and

line 151 removes a before dysregulation

line 153 adds a comma before and

line 154 changes significant to significantly

line 156 adds a comma before and                                  

line 158 changes general to generally

line 159 changes significant to significantly

line 160 adds a comma before and

line 168 adds the before figure

Line 172 changes are to is

Line 177 adds in before downregulated

Line 178 changes remain to remains

Line 179 adds in after resulted

Line 180 adds a before nasopharyngeal  

line 186  adds a before nasopharyngeal  

line 192 changes disease to diseases

line 195 changes significantly increase of to a significant increase in                                                                         

line 198 adds a or the before cellular

line 201 changes with respect to to concerning

line 208 changes analysed to analyzed

Line 211 changes to to with

line 213 change are to is

line 215 changes adult onset to adult-onset

line 222 changes mechanism to mechanisms

Line 219 changes haemorrhagic to hemorrhagic

line 228 add a comma after date

Line 232 adds a comma after study

line 234 adds a comma before  and

line 235 changes respect to concerning

line 235 add the before long-COVID

Line 247 adds the before acute

line 251 changes to to with

Line 253 removes that before are

Line 256 changes here reported to here-reported

Line 257 adds the before RBC

Line 258 changes of to for

Line 259 changes enzyme to enzymes

Line 260 changes of to in

Line 261 changes shown to have shown or showed

line 262 adds a before protective

line 263 adds a comma before or

Line 265 changes of to for

line 266 adds a comma before and

line 267 change stress activated to stress-activated

line 270 adds a comma after patients

line 271 adds a comma before and

line 276 removes the space after Tf

line 283 adds a comma before and

line 287 changes is to are and adds the before management

line 294 remove in before long

Line 296 adds a before significant

Line 297 removes that before seems

line 298 changes that to which

Line 300 changes candidate to candidates

Line 301 adds the before progression

line 302 changes  carrying out of lymphocytes to carry out lymphocytes

Line 304 changes others to other

line 309 adds a comma before and

Line 310 removes be before propagate

line 315 adds the before severity

Line 316 removes an before increased

line 324 remove been before emerged

something wrong between lines 325 to 330 in colors or font

line 326 adds a comma after iron

line 326 adds a or the before cellular

line 335 removes a before LCN2

Line 337 changes induce to induces and remove the before other

line 339 changes increase the iron to increasing iron

line 340 changes of to in

line 341 add a comma before and

line 342 removes a before posttranslational

345 changes          avaibility to availability              

Line 347 changes modulates to modulate

line 355 changes iron related to iron-related

Line 357 changes cohort to cohorts

Line 358 changes analysis to analyses         

line 361 removes the before COVID-19

Line 369 changes (volunteers donors) to (volunteer donors) or (volunteers donor)

Line 372 changes of of to of

Line 373 add a comma before and

line 381 changes disease to diseases

Line 383 changes diamino to diamine

Line 383 changes tetracetic to tetra acetic

line 389 changes fluorescence activated to fluorescence-activated

line 390 changes (FACSAria III, BD Bioscences) to (FACSAria III, BD Biosciences)

Line 393 changes increase in protein to an increase in the protein

line 396 changes triton X to Triton X

line 405 add an before alkylating

Line 406 changes analysed to analyzed

Line 406 changes Chromatography-Tandem to  

Line 415 changes 1 minutes to 1 minute

line 416 changes maintaining to maintains and 1 minutes to 1 minute

Line 424 adds the before protein and changes level to levels

line 425 changes analysed to analyzed

line 428 changes Intensity based to Intensity-based

line 444 changes separated in 12 to separated using 12

line 444 adds a comma before or

lines 447, 448, 449, and line 450 change signalling to signaling

Line 449 changes ant-LCN2 to anti-LCN2

Line 450 add a comma before and

line 462 changes were analysed to was analyzed

line 463 adds a comma before and

line 464 changes were to was

Line 471 adds a before reference

Line 447 adds a comma before and also changes heathy to healthy

Line 481 changes were to was

Line 485 changes were to was

line 487 changes colour to color

line 491 changes group to groups

line 493 adds the before mean

Author Response

Response to Reviewer 2 Comments

Dear Reviewer,

We thank you for your constructive criticisms and suggestions, which allowed us to significantly improve the manuscript. We hope the manuscript can be accepted in this revised form.

Changes responding to all the Reviewers are yellow highlighted in the revised text.

Please find here following our point-by-point answers.

Point 1:

The whole manuscript is mixed between American English and British English, the authors even use British English or American English. For consistency, consider replacing it with the American English spelling.

Response 1:

We would like to thank the Reviewer for his/her accurate analysis, corrections, and suggestions. We considered to replace American English spelling.

Point 2:

The English writing should be further improved, as there are many grammatical or typing errors. It is suggested to ask a native speaker to polish it.

Response 2:

We have chosen to use a paid editing service for the extensive English revisions required and have obtained a confirmation certificate.

Point 3:

The bibliographic references should be revised according to the journal's instructions

Response 3:

We have revised the bibliographic references according to the journal's instructions.

Minor points

We thank the Reviewer for the suggested minors point that were taken into account in the revised manuscript.

Reviewer 3 Report

This work is e interesting and comprehensive. However, there are some concerns as follows: What was the basis of the study grouping? What is the exact difference between hospitalized patients and post-acute ones? What is the power of study? In the third group, there are 10 patients. Was it statistically sufficient? Please declare all including and excluding criteria Please change group naming to better keywords that show the group characteristics. Why QPCR was done just for COVID-19 group? how about long covid? How long past for the recovered COVID-19 group? were they the same? In the supplementary file please correct "Ceruloplasmina" and "Transferrina"

Author Response

Response to Reviewer 3 Comments

Dear Reviewer,

We thank you for your constructive criticisms and suggestions, which allowed us to significantly improve the manuscript. We hope the manuscript can be accepted in this revised form.

Changes responding to all the Reviewers are yellow highlighted in the revised text.

Please find here following our point-by-point answers.

Point 1:

What was the basis of the study grouping?

Response 1:

We thank the Reviewer for his/her comment, and we consequently improved the methods section describing more into details experimental and study design, as well as the grouping, and the including/excluding criteria (line 472-483, 485-487 page 14). Consequently, we implemented the paragraph called “4.1. Patients and sample information” in materials and methods section.

Point 2.

What is the exact difference between hospitalized patients and post-acute ones?

Response 2.

We thank the Reviewer for the opportunity to better clarify this point. The difference between COVID-19 hospitalized patients (called COVID-19) and patients with post-acute infection symptoms (called long-COVID) was that the first canonically have an active infection with the well-described symptoms, while the second, as recently described in literature (we add further bibliography in the revised manuscript, [20-22, 25, 29]) share the following common long-term (more than 3 months after the SARS-CoV-2 infection resolution) symptoms; chronic fatigue, breathlessness, cardiovascular abnormalities, neurocognitive impairments, anxiety, and depression.

According to the Reviewer suggestion, we added a description clarifying this point in the paragraph 2.1 Clinical characteristics of COVID-19 and long-COVID patients in Results Section (lines 137-141, page 4).

Point 3.

What is the power of study?

Response 3.

We thank the Reviewer for the question and in accordance with his/her comment Power calculations were performed using G*Power version 3.1. The achieved power for LTB4 and LCN2 analysis were 0.79 and 0.82 with an effect size d = 1.17 and d = 1.13.

We included the power of the study in the paragraph 2.5. Increased LTB4 and LCN2 plasma levels in COVID-19 and long-COVID-19 patients in Results section (line 261, page 9).

Point 4.

In the third group, there are 10 patients. Was it statistically sufficient?

Response 4.

We agree with the Reviewer that the long-COVID group is limited in number, indeed we already declared in the discussion section the following sentence: “A limitation of this study is the number of subjects included in the two cohorts of patients, particularly for long-COVID and further analyses are required in a larger cohort of patients in order to better validate LTB4 and LCN2 as a potential biomarker for COVID-19 and long-COVID”. To further underline this concept, we improved this description in the discussion section (lines 448-449, page 13).

Point 5.

Please declare all including and excluding criteria

Response 5.

We thanks the Reviewer for this comment, we added the including and excluding criteria in revised manuscript, in the paragraph called “4.1. Patients and sample information” in materials and methods section (lines 472-483, 485-487, page 14).

Point 6.

Please change group naming to better keywords that show the group characteristics.

Response 6.

We hope that after the changes in paragraphs called 2.1 Clinical characteristics of COVID-19 and long-COVID patients and 4.1. Patients and sample information”, suggested by Reviewer, describing in a clearer and more detailed way the characteristics of COVID-19 and long-COVID groups. Furthermore, the grouping names were thus not changes, better justifying them with additional bibliography and taking into account they are also named in the same manner in the Special Issue Information/description.

Point 7.

Why QPCR was done just for COVID-19 group? how about long covid?

Response 7.

We agree with the point raised by the Reviewer. Actually, we analyzed the gene expression levels of 5-LOX using RNA extracted from the nasopharyngeal swabs, samples collected for SARS-CoV-2 diagnosis. Unfortunately, we did not collect the same samples for the long-COVID group to analyze the 5-LOX gene expression levels. However, we do not consider these data strictly necessary, so we moved the gene expression data in the Supplementary section, as Supplementary Figure 2.

Point 8.

How long past for the recovered COVID-19 group? were they the same?

Response 8.

We thank the Reviewer for the question. The cohort of long-COVID group included long-COVID patients, from 3-6 months post-infection. We added this information on revised manuscript in the paragraph called “4.1. Patients and sample information” in materials and methods section (lines 472-483, 485-487, page 14). About lane 470, we changed the trivial error "recovered COVID-19" with  "hospitalized COVID-19". We are sorry for the mistake.

Point 9.

In the supplementary file please correct "Ceruloplasmina" and "Transferrina"

Response 9.

We are sorry for the mistake. Names are now correct.

Reviewer 4 Report

This study is aimed at identifying possible mechanisms and biomarkers associated to iron metabolism in COVID19 and long-COVID patients. The authors performed proteomic analysis in lymphocytes and identified LTB4 and LCN2 as possible biomarkers for COVID19 and long-COVID. Moreover, the 5-lox gene resulted up-regulated in COVID19 patients compared to control group, conversely 5-LOX protein was down-regulated. Overall, the work is interesting even if some data presentations could be strengthened by addressing some points:

-    The authors reported in a previous work (Dufrusine B et al 2019) the iron-mediated 5-LOX activation in macrophages and its correlation with the inflammatory processes. How could the authors explain the down-regulation of 5-LOX protein in COVID19 patients compared to healthy donors in relation to inflammatory response observed during the infection? This aspect should be better discussed in paragraph 2.4.

-  Although the reported data are interesting and suggest a role for iron overload in COVID19, in vitro data on the modulation of LOX5 at gene and protein expression following SARS-CoV-2 infection could clarify the regulatory mechanism and its association with the inflammatory response during the infection.

-  The oxidative stress is a main feature of different respiratory viral infections including SARS-CoV-2. In this context, different works reported data about the inhibition of antioxidant responses during the infection that should be reported in lines 260-263. Concerning the redox dyshomeostasis, oxidative stress markers (e.g. ROS production) should be better characterized in the COVID19 and long-COVID samples.

Author Response

Response to Reviewer 4 Comments

Author Response

Dear Reviewer,

We thank you for your constructive criticisms and suggestions, which allowed us to significantly improve the manuscript. We hope the manuscript can be accepted in this revised form.

Changes responding to all the Reviewers are yellow highlighted in the revised text.

Please find here following our point-by-point answers

Point 1.

The authors reported in a previous work (Dufrusine B et al 2019) the iron-mediated 5-LOX activation in macrophages and its correlation with the inflammatory processes. How could the authors explain the down-regulation of 5-LOX protein in COVID19 patients compared to healthy donors in relation to inflammatory response observed during the infection? This aspect should be better discussed in paragraph 2.4.

Response 1.

We thank the Reviewer for giving as the possibility to improve the discussion of this relevant result concerning the 5-LOX downregulation in both COVID-19 and long-COVID patients versus controls. This question prompted us to better analyze and discuss the role of specific cytokines in the regulation of 5-LOX levels (lines 379-387, page 11).

Point 2.

Although the reported data are interesting and suggest a role for iron overload in COVID19, in vitro data on the modulation of LOX5 at gene and protein expression following SARS-CoV-2 infection could clarify the regulatory mechanism and its association with the inflammatory response during the infection.

Response 2.

We agree with Reviewer regarding this intriguing observation. Future Studies are planned to address in vitro analyses to further validate the molecular mechanism.

Point 3.

The oxidative stress is a main feature of different respiratory viral infections including SARS-CoV-2. In this context, different works reported data about the inhibition of antioxidant responses during the infection that should be reported in lines 260-263. Concerning the redox dyshomeostasis, oxidative stress markers (e.g. ROS production) should be better characterized in the COVID19 and long-COVID samples.

Response 3.

We agree with the Reviewer observation. Concerning the oxidative-related markers, we investigated only the SOD1 protein levels in PBMCs. I would be interesting measuring also the ROS levels in plasma samples, but in this case we would have added chemical agents to prevent auto-oxidation during storage, as recommended for measuring ROS in biofluids, a common problem in all these analyses leading in some cases to contrasting results (Halliwell, B. & Gutteridge J. M. C., Free Radicals in Biology and Medicine 5thend, 2015; Liu, W. et al. Ex vivo oxidation in tissue and plasma assays of hydroxyoctadecadienoates: Z,E/E,E stereoisomer ratios. 2010; Murphy et al., Guidelines for measuring reactive oxygen species and oxidative damage in cells and in vivo, 2022) and for this reason this was out of the main objectives of our research. However, in agreement with your comment, we changed the title of the manuscript from “Iron and Redox dyshomeostasis in COVID-19: biomarkers reveal a functional link to 5-lipoxygenase activation” to “Iron dyshomeostasis in COVID-19: biomarkers reveal a functional link to 5-lipoxygenase activation”.

Moreover, following the Reviewer suggestions, we added a Supplementary Figure 3 describing the activation of pathways leading to redox dyshomeostasis obtained by IPA analysis performed on our proteomic data. Furthermore, we added the sentence and reported further studies concerning oxidant response during COVID-19 and long-COVID in revised manuscript in discussion section (line 335-343, page 10).

Round 2

Reviewer 3 Report

There is a very number of references. If the editorial team of the journal is ok with this, the revised version is now acceptable.